# Trust Relationship with Suppliers, Collaborative Action, and Manufacturer Resilience in the COVID-19 Crisis

**DOI:** 10.3390/bs13010033

**Published:** 2022-12-30

**Authors:** Jianhua Yang, Yuying Liu, Moustafa Mohamed Nazief Haggag Kotb Kholaif

**Affiliations:** School of Economics and Management, University of Science and Technology Beijing, Beijing 100083, China

**Keywords:** trust, collaborative action, manufacturer, resilience, behavioral operations management

## Abstract

Frequent emergencies highlight the importance of corporate resilience, and relationship governance behaviors have received much attention for improving firms’ resilience. This study uses structural equation modeling to test hypotheses based on trust theory and synergy theory. It investigates the impact of the behavioral operations management approach of collaborative action on manufacturers’ resilience in Chinese manufacturing firms during the COVID-19 pandemic. Environmental uncertainty is used as a moderating variable to explore its role between the trust relationship with suppliers and collaborative action, which are two types of relationship management behaviors. The results show that collaborative action positively affects each of the three stages of manufacturer resilience. Moreover, collaborative action mediates between the trust relationship with suppliers and the different dimensions of manufacturer resilience. Environmental uncertainty does not moderate the facilitative effect of the trust relationship with suppliers on collaborative action. This study examines the relational governance behavior of firms in the context of a particular Chinese culture to build a bridge between relational governance behavioral science and firm resilience management. This study tries to provide inspiration for managers to improve the risk management ability of manufacturing enterprises by using relationship management theory and behavioral science.

## 1. Introduction

Frequent emergencies in recent years, such as the COVID-19 pandemic, have caused supply chain inefficiencies and late deliveries [1,2,3]. Improving the resilience of companies in the supply chain is a topic that deserves extra attention at present [4,5,6]. The study of resilience has its origins in social psychology [7], and the multifaceted and multidisciplinary idea of supply chain resilience has gained substantial interest from the operations management community [8]. Resilience is a multidisciplinary concept that refers to the ability to adapt to changes and cope with emergencies while maintaining the basic functions and structure of the system [9]. The ability of supply chain operators to absorb, recover from, and react to interruptions of varying durations, consequences, and probabilities are essential to the company’s sustainability [2]. Previous studies have mostly focused on supply chain resilience and disruptions [10,11,12]. However, the previous literature rarely distinguished between specific research subjects within supply chain members, such as manufacturers or customers [13]. The literature on supply chain resilience has grown exponentially, but research focusing on manufacturer resilience is still limited [14]. Moreover, there are few empirical methodologies. As different supply chain members have various operational characteristics, this paper explicitly investigates the resilience of manufacturers and attempts to provide new perspectives on the operational management of manufacturing firms.

The importance of supplier management was highlighted during the pandemic. Contractual governance could not resolve issues between manufacturing companies and suppliers quickly and efficiently in emergencies, and relationship governance was urgently needed to compensate for the lack of contractual governance. Scholars have already studied enterprise risk management by examining buyer–supplier relationships [15]. However, due to the specificity of Chinese culture and the unprecedented nature of the COVID-19 pandemic, previous contractual governance still needs to address the problems of Chinese manufacturing companies in the context of the COVID-19 pandemic. In the event of an emergency, such as a natural catastrophe or the COVID-19 pandemic, trust and collaboration amongst supply chain partners are crucial for establishing resilience, minimizing the risk of disruption, and organizational development [16]. Due to geographical proximity and close management ties, as well as being based on trust, cooperative enterprises form formal (contracts) or informal patterns of cooperation (communication and behavior). The importance of relationships in synergy is confirmed by the fact that a sufficiently high-quality relationship reduces opportunistic behavior and minimizes the risk of cooperation [17]. Chinese companies’ managers are influenced by the subtle influence of traditional Confucian culture, which affects corporate decisions and behavior [18]. In operations management, behavioral factors affect partnership, and Western relationship governance theory cannot solve China’s problems. This study considers Chinese Confucian culture, focuses on significant national demands, and addresses Chinese problems in a novel Chinese way. 

Studies have shown that relationship governance affects firm resilience in supply chains [16]. According to supplier relationship management theory, collaboration is ideal for company–supplier relationships [19]. Research has shown that increased supplier relationship capital reduces buyer performance [20]. However, studies also show that collaboration can replace or reduce audit and monitoring activities due to supplier development strategies. The move towards a lean supply chain can be exacerbated by the propagation of disruption by companies aiming to reduce storage costs through collaborative management [21]. Forming a partnership with suppliers facilitates responsible and profitable supply chains and improves resilience. When collaborating, companies always think the same; one party’s ideas are superfluous, which depends on the strength and power of the two. The process of collaborative action (COL) is complex [22]. According to the synergy theory, long-term partnerships between companies and their stakeholders are achieved by building trust and commitment [23]. Trust and collaboration-based norms and relationship embedding lead to the development and integration of resources [20]. Although researchers studied the effect of relationship management on supply chain resilience [6], the current research on the impact of collaborative action between manufacturers and suppliers on resilience is limited. Existing research needs an exploration of the impact of collaborative action on manufacturing firms’ resilience in emergencies. Supply chain synergies aim for “win–win” relationships, aligning interests and striving for coexistence. The theory of supply chain synergy proposes that the method for beginning synergies is dependent on the motivation of at least one enterprise and the contractually outlined standards of cooperation [17]. Partners also use risk-averse strategies in the cooperation game, which can affect resilience [24]. However, other scholars argue that collaborative action is essential for resilience, and collaboration can produce visibility, agility, and flexibility [25]. The role of joint action in the resilience of manufacturing companies is a controversial topic worth investigating.

The significance of this study is reinforced by the following. Firstly, this study combines theory with practice, based on supplier management theory and collaborative risk management theory, coupled with field research in manufacturing companies. This study overcame many difficulties in collecting data to reflect the actual corporate situation during the pandemic, and the data are invaluable. Secondly, this study integrates knowledge across disciplines, builds a bridge between the two fields of supplier relationship management and corporate risk management [3,23], and enriches the methods and ideas for improving enterprise resilience, which has theoretical value. This study is the first attempt to apply relational governance theory in the context of Chinese culture to solve Chinese problems [26]. The epidemic raised awareness that the zero-stock strategy of pursuing efficiency is not fully applicable in emergencies under unexpected events, and that the objectives of zero-stock management are not the same as those of corporate emergency management [27]. Manufacturing companies are now in dire need of resilience enhancement programs, making this study of practical relevance. In practice, scholars claim that during emergencies, it is vital that partners cooperate to reduce the risk of interruptions through communication, and trust one another, thereby increasing resilience [28]. Trust and collaboration between partners will minimize the adverse effects of manufacturers’ operational disruptions [29]. The impact of trust on resilience has been demonstrated [28]. However, there needs to be more investigation into the relationship between the trust relationship with suppliers (TRS), collaborative action, and firm resilience in emergencies. According to Srinivasan and Swink [30], organizational flexibility enables businesses to adapt their strategies in response to environmental unpredictability. Therefore, the motivation for this study is to answer the following relevant questions to fill in the gaps. 

RQ1: Does collaborative action promote manufacturer resilience? Which dimensions of manufacturing firm resilience are affected by collaborative action? 

RQ2: Does collaborative action act as a mediator between TRS and manufacturer resilience?

RQ3: Will different environmental uncertainties affect the promotional effect of TRS on collaborative action? 

The innovation of this paper stems from the theoretical and practical implications. Firstly, this study adds to the gap that links collaborative action relationship governance approaches to manufacturer resilience [3,23]. This study attempts to verify the impact of collaborative action on manufacturer resilience during unexpected events, where opportunism and gaming co-exist rather than not, under normal circumstances [31]. Moreover, this paper combines Chinese and Western cultures and theories, such as the Western contract system [20], military knowledge from the ancient Chinese scholar Sun Tzu’s *Art of War* [26], knowledge of disaster science, and emergency management theory [10,32]. This study opens the door to applying relationship governance approaches to improve manufacturer resilience, and outlines the differences in the impact of trust on collaborative action as a form of relational governance under different environmental uncertainties. Thirdly, this study clarifies the concept of collaborative action between manufacturing companies and suppliers in an emergency [33]. Theoretically, this study complements collaborative risk management theory [34], while practically, it can guide the operational management of manufacturing companies [28,35]. Practically, our paper provides novel insights into how manufacturers can improve resilience and risk tolerance during the COVID-19 pandemic.

## 2. Literature Review and Research Hypotheses

### 2.1. TRS and Collaborative Action

The concept of “synergy” derives from the theory of synergetics in the natural sciences, developed in the 1970s by the prominent German physicist Hermann Haken [36]. The concept of synergy has progressively gained acceptance in the business management community as the study of systems science has matured. Following the study of Scholten and Schilder [37], we utilize the synergy theory to describe collaborative action. Collaboration is a strategy to counteract supply chain disruptions and improve firm competitiveness and business performance [38]. Collaborative action is characterized by “shared planning” and “joint problem solving”. Joint planning refers to the degree to which suppliers and manufacturers agree before unforeseen events, necessary duties, and assignment procedures in the process of their collaboration, which is a kind of prediction management. Joint problem solving refers to the extent to which problems encountered by suppliers and manufacturers are effectively resolved, which is a reactive governance approach [39]. Supply chain collaboration has become a strategic issue in achieving the goal of companies’ resilient and sustainable growth [40]. 

Collaboration between manufacturers and suppliers includes developing pertinent plans, proposed solutions, and procedures for the supply chain’s ultimate strategy. Moreover, they need to jointly assume relevant responsibilities so that the cooperative enterprises can coordinate, and each link can be seamlessly connected. Based on the synergy theory, it is vital to highlight that supply chain synergy is a collection of procedures and includes several sophisticated ideas and abilities that make it possible. In addition to information sharing and integration, supply chain synergy includes tactical decision making in collaborative planning, predictive distribution, and product design. Manufacturers and suppliers employ information technology and other management strategies to integrate their resources and execute dynamic management of synchronized inter-company activities and seamless interfaces to block, attenuate, and accentuate unfavorable developments [34]. 

Trust is an implicit condition for collaboration [41], and an essential prerequisite and management mechanism for building relationship capital with suppliers, enabling supply chain partners to focus on the long-term benefits of the relationship, thereby increasing competitiveness [42]. In fact, “cooperation and trust” are inseparable. According to Narayanan et al. [43], buyer–supplier relationships can be positive, negative, or neutral, depending on the level of trust. Wagner et al. [44] found that trust during project collaboration substantially impacts the buyer–supplier relationship’s future. Campos et al. [41] emphasize trust as a factor of firm–network collaboration.

Trust and collaboration have been linked in new ways by the trust theory, which describes how relationships with suppliers facilitates collaboration. Moreover, synergy theory offers new perspectives on supplier relationship governance in manufacturing enterprises during a pandemic. In China, a cooperative connection based on mutual trust between enterprises, as a significant societal and cultural element, has a substantial impact on companies’ business decisions and collaborative actions [45]. However, the interaction effect between manufacturer–supplier collaborative action and manufacturer resilience in supply chain relationship management has received scant attention. The trust relationship between manufacturers and suppliers has been described as one in which firms expect their partners to undertake specific activities to serve their interests, regardless of the firm’s ability to monitor its partners [46]. Eastern cultures have opened the door to traditional ideas of collaboration, which say that trust formed by earlier cooperation, motivated by a common interest, can serve as a good starting point for future cooperation.

Consequently, trust plays a vital role in shaping new supplier–manufacturer relationships when supply chain collaboration happens [47], for example, in joint problem solving. Pomponi et al. [48] suggested that collaboration-driven trust is the highest level of trust in collaboration and that a minimum level of mutual trust is a prerequisite for collaboration to begin. In the Chinese environment, trust between sourcing managers and supplier representatives is often more secure than a signed contract [49]. Several studies have suggested that supply chain partnerships can be maintained through the deployment of trust [50]. Working partners with high trust are less hesitant to provide complete information and trust the information they receive, resulting in a higher desire to act. High levels of trust, in particular, offer incentives for open communication and willingness to take risks amongst partner enterprises in a manufacturer–supplier relationship [51]. Consequently, a trustworthy transactional environment between partners may be crucial in fostering collaboration in supply chains. Trust is a prerequisite for creating exceptional supply chain collaboration [52]. Thus, we can develop the first hypothesis as follows: 

**Hypothesis 1.** *TRS has a significant positive effect on collaborative action*.

### 2.2. Collaborative Action and Manufacturer Resilience

According to Qi and Hui [53], the typical phases of emergency management are divided into four stages: prevention, preparation, response, and recovery (PPRR). This is the PPRR theory, which is widely used in crisis management. Specifically, the prevention phase consists of assessing the crisis environment, identifying key factors that may lead to a crisis, and addressing them as early as possible. The preparation phase involves the development of emergency plans, which allow for the early identification of possible crisis events. The response phase, which begins with the occurrence of an emergency, comprises timely and effective measures to stop the crisis and prevent a recurrence of the disaster. The recovery phase is about restoring the change and order caused by the emergency. Ponomarov and Holcomb [54] defined the essence of supply chain emergency resilience as three phases: preparation, response, and recovery. During the preparation time, supply chain partnerships are developed with an emphasis on efficiency, risk aversion, reliability, contingency systems and procedures, and informatics updates. The reaction phase is marked by adaptability, timeliness, information sharing, and shared risk. The recovery period requires attention to recovery cycles, delivery capabilities, warehousing efficiency, customer service levels, and highly integrated system processes. The components of resilience are defined in terms of the capabilities of the three phases mentioned above: preparedness (PPA), responsiveness (RPA), and recovery capabilities (RCA). Therefore, in this study, manufacturer preparedness under emergencies refers to the ex ante allocation of resources to detect sources of risk in order to prevent supply chain disruptions and maintain the originally planned level of operations. Responsiveness refers to the interaction between resources to help firms respond to supply chain disruptions and to share relevant information quickly with other member firms. Recovery capability refers to the ability of companies to reprogram and coordinate supply chain resources and return to desired production levels promptly.

#### Collaborative Action and Preparedness

Formentini and Romano [55] defined supply chain collaboration as the coordination of several participants to gain a competitive edge by facilitating communication, making collaborative choices, and exchanging advantages. This leads to more customer service profitability than if they had worked alone. Much previous research implies that collaboration in the supply chain can help improve company productivity in various ways [56]. According to Um and Kim [57], supply chain collaboration improves firm performance and lower transaction costs. However, other academics claim that implementing supply chain collaboration in the actual world does not provide firms with the desired benefits [58]. Schulz and Blecken [59] confirmed that mutual distrust and a lack of transparency are the main barriers to a collaborative approach in post-disaster relief logistics. Previous relationship literature implied that when a company invests in building relationships with suppliers, it is more likely to work with them or integrate better, as solid ties allow for regular contact and information exchange [60]. Thus, we can develop the next hypothesis as follows:

**Hypothesis 2a (H2a).** *Collaborative action has a significant positive effect on preparedness*.

Mcevily and Marcus [61] argued that joint problem solving between firms could facilitate knowledge transfer between firms, and reduce opportunistic behavior through frequent interactions. Joint problem solving between cooperating firms facilitates the learning of experiences and knowledge in companies, which promotes technological innovation and thus increases the resilience of manufacturing firms [33]. In addition, the buying corporation’s willingness to collaborate is typically greater than the supplier’s need. Typically, the purchaser firm takes the initiative in the collaboration, which is expected to facilitate the creation of synergies. Based on the above, we propose the following hypothesis:

**Hypothesis 2b (H2b).** *Collaborative action mediates the relationship among TRS and preparedness*.

Collaborative planning, forecasting, and replenishment can increase a company’s responsiveness by coordinating supply chain operations such as forecasting, manufacturing, and sourcing [62]. Moreover, collaboration improves information exchange and communication between manufacturers and suppliers, minimizing information disparities and fostering dependable relationships [63]. Supply chains are founded on collaborative behavior, in which a joint decision-making procedure is formed to fulfill the parties’ shared objectives [64]. Collaborative behavior enables manufacturers and suppliers to comprehensively understand future demand, establish a realistic strategy to satisfy that need, and systematically coordinate all connected actions [55]. Therefore, collaborative action drives successful supply chain procedures. Social exchange theory (SET) contends that a supplier may contribute to its manufacturer through collaborative practices and collaborations. According to the theory of collaborative risk management, suppliers and manufacturers collaborate to analyze risk factors and take appropriate measures to avoid and prevent risks. Partners can minimize risk losses, enhance stability, protect the enterprise’s interests, and operate efficiently together. Therefore, the researchers propose the following hypothesis:

**Hypothesis 3a (H3a).** *Collaborative action has a significant positive effect on responsiveness*.

Collaboration in supply chains encompasses the mutual support of firms, such as manufacturer–supplier partnerships. Closer relationships between suppliers and manufacturers help to share risks, access complementary resources, and improve productivity and supply chain resilience [44]. Business interaction is an external action undertaken by purchasers and vendors that is influenced by external conditions laden with uncertainty [65]. Consequently, firms that interact based on trust perform better. Specific collaborative activities between manufacturers and suppliers, such as collaborative communication and mutual relationship efforts, can improve a company’s resilience during emergencies by increasing visibility, speed, and flexibility [66]. Thus, the researchers develop the following hypothesis:

**Hypothesis 3b (H3b).** *Collaborative action mediates the relationship among TRS and responsiveness*.

Occasionally, buyer firms function better than various suppliers, indicating that organizations in the supply chain must emphasize the long-term synergy functioning of supply chain cooperation above the pursuit of speedy short-term profitability. Manufacturers who have a positive relationship with their suppliers will have a more stable operation and be able to rapidly resume production after an emergency. Recognizing the greater benefits offered by competitive suppliers, more and more manufacturers trust suppliers and are willing to work closely with their suppliers [67]. Closer collaborative action between suppliers and manufacturers helps to share risks, access complementary resources, and increase productivity and resilience over time [37,55]. Therefore, we propose the following hypothesis:

**Hypothesis 4a (H4a).** *Collaborative action has a significant positive effect on recovery capability*.

Gao et al. [68] demonstrated that alliance partners can share needs and behavioral expectations through collaborative action and that, over time, collaborative action can result in shared behavioral norms. This shows that as confidence between partners builds by joint action, they are better equipped to resolve difficulties through tacit agreement, reduce opportunistic behavior on both sides, and restore and maintain corporate stability after crises. The content or frequency of communication shared in collaborative action depends on the closeness of the collaborative action, which is essential for improving the company’s capabilities [47]. Based on the previous literature, we propose the following hypothesis:

**Hypothesis 4b (H4b).** *Collaborative action mediates the relationship among TRS and recovery capability*.

### 2.3. The Moderating Role of Environmental Uncertainty

Partnerships have been defined as cultural factors that impact business decisions and the behavior of manufacturing and supplier companies as an informal form of relationship governance [35]. Trusting relationships foster buyer–supplier cooperative synergy, whereas uncertainty can be detrimental to long-term collaboration [69]. According to Cheng et al. [70], excellent partnerships between enterprises and suppliers promote corporate information sharing. Considering these multiple perspectives, we argue that there is a need to investigate further the role of trusting relationships, rooted in the uncertain environment during emergencies, on collaborative actions between Chinese manufacturing and suppliers. Using social exchange theory, Yang et al. [50] uncovered the determinants of relationship stability in supply chain coalitions. The existence of trust enhances the work atmosphere by boosting contractual reliability, encouraging cooperation, and reducing risk and ambiguity. Collaboration between cooperators is influenced by trust and interdependence. As the cooperators are willing to work jointly, Collaboration will lead to more stable transactions and reduced uncertainty in the market environment [71]. Several studies have also demonstrated that perceptions of exchange relationships, such as trust, are crucial to the success of supply chain cooperation choices [72].

Although manufacturers and suppliers can bind each other in their cooperation when circumstances change, both sides may be unwilling to make an effort to retain this synergistic relationship. They may instead seek out other partners [73]. This is because this prevents conflict and has the potential to yield additional benefits. Organizations may prioritize the development of competitive advantages over collaborative advantages for their own benefit [74]. Lambert et al. [75] claimed that strong drivers (such as cost efficiency and market advantage) and a supportive environment foster successful collaboration between enterprises. Moreover, supplier–manufacturer collaboration necessitates a complicated social process that the environment may heavily influence. In the absence of significant drivers, environmental circumstances can substantially impact the formation and maintenance of cooperation. In favorable circumstances where individuals are not gloomy, promises are typically kept. Despite their willingness to act logically in business situations, managers are limited in every situation requiring knowledge and communication. The more uncertain the environment, the greater the difficulties. This may result in undesired behaviors such as lying, cheating, and agreement violation [76]. Environmental uncertainty (ENU) is an external element that impedes trust formation. It is crucial to recognize that the continual evaluation of the external environment and the role of its influence is predictable, even when the partnership is viewed as a primarily long-term endeavor [77]. From this perspective, the researchers develop the following hypothesis and the theoretical model(see Figure 1):

**Hypothesis 5 (H5).** *Environmental uncertainty moderates the relationship between TRS and collaborative action*.

## 3. Methodolody

### 3.1. Data Collection

The sample of Chinese manufacturing companies in this study was drawn from the following sources: members of the China Association for Quality, members of the China Federation of Logistics and Purchasing, manufacturing companies with which the research team collaborated on research projects, members of independent director training courses of the listed companies, and students of MBA classes in Chinese universities. The China Association for Quality has more than 1000 members, including Baosteel, Haier, Lenovo, FAW, Shanghai Mitsubishi, Wuliangye, and Intel (Shanghai). The number of members in the China Federation of Logistics and Purchasing is more than 400. The study population came from the above-mentioned organizations located in various cities in China. The population in this study comprised nearly 1600 individuals. The population surveyed in this study comprised managers and independent directors of manufacturing companies. A snowball sampling method was used to distribute the questionnaires, where we found managers of manufacturing companies and asked them to answer the questionnaires, and then send the questionnaires to other manufacturers they knew to complete the questionnaires. The data were collected between January 2022 and September 2022. Our research team has a long-term cooperation with manufacturing companies, but due to the COVID-19 pandemic and strict lockdown policies, the project was suspended for some time. In order to collect more data, the data collection for this study therefore took longer than other studies that were not affected by the pandemic. Part of the work was carried out online, and we also met managers of new manufacturing companies during our work and distributed questionnaires to them in person. Before the questionnaire was formally distributed, we consulted with 12 academics in economics and management-related professions and managers working in manufacturing companies to ensure that the questionnaire was clear and understandable. The discussion of this study involved four PhD students, four managers of manufacturing companies, two independent directors and two university teachers. We have close working links with these people. As this investigation was undertaken during the ongoing COVID-19 pandemic, a small number of companies were visited, and paper copies of the questionnaires were distributed on site, while more were obtained online via WeChat and email.

The size of the sample in this study was 556, and a total of 382 effective questionnaires were returned. The response rate was 68.71%. To ensure the questionnaire would be responded to, we explained the significance of this study and offered consultancy services to companies which we had cooperated with in the projects. After giving the managers a paper version of the questionnaire on site, the directors gave it to the managers of their departments and asked them to complete it carefully. Meanwhile, these managers sent the questionnaire to their well-connected workmates. The research team has a good network of cooperation with the manufacturing companies under investigation. We used the bootstrapping method with 5000 replications by SPSS and AMOS software to estimate our model [78,79]. Companies are located in China’s first-tier cities, such as Beijing and Shenzhen. Cities of the second tier include Qingdao and Tianjin, and third-tier cities include Hohhot and Shijiazhuang. As shown in Table 1, the sample contains companies of different natures and covers different industries. The surveyed companies are of different sizes to ensure that the sample is representative.

### 3.2. Nonresponse Bias and Common Method Bias Analysis

The following were performed in order to prevent no-response bias that would misevaluate the construct variables. First, it was explained to the respondents in advance that the questionnaire would be used for scientific research only, and that there would be no attempt to commercially benefit from it. Moreover, the questionnaires completed by the respondents were anonymous. Second, respondents were informed that there was no correct response and that it was crucial to choose the option that corresponded with the company’s real circumstances. This ensured that the answers were factually correct and had no subjective bias so they could self-report [80]. Additionally, the context and relevance of this study were explained to the respondents so they could fully understand the items in the questionnaire. Finally, the researcher made a return visit within a week of the distribution of the questionnaire to remind respondents who had forgotten to complete the questionnaire to respond immediately [81]. All the above methods mitigated the effect of non-response bias on the study results.

In terms of preventing common method bias, we made the following efforts. Firstly, to ensure the objectivity of the variables measured, information was obtained from various sources. Each variable was divided into different subsections in the questionnaire so that respondents could clearly distinguish between the different variables [82]. In addition, we have further implemented Harman’s one-factor test, a traditional method used to test for common method bias [83]. Table 2 demonstrates the test results [84]. We compared the model fit of the original model with that of the single-factor model and the standard method factor model. Compared to the original model, the model fit was worse for the one-factor model, whereas there was no substantial difference for the common method factor model. Therefore, these can demonstrate that this study has no serious common method bias [85].

### 3.3. Measures of Constructs

The questionnaire items used in this study were derived from validated scales in the existing literature. They were adapted based on expert opinion during the research to ensure that the companies’ managers understood the questions well (see Table 3). The respondents were requested to fill out a seven-point Likert scale, with 1 indicating “seriously disagree” and 7 indicating “strongly agree”.

#### 3.3.1. Trust Relationship with Suppliers (TRS)

The independent variable, TRS, consists of seven items [28,86,87]. The items show the extent to which the respondent thinks the manufacturing companies they work for trust in the competence and attitude of their suppliers. Specifically, the items on the scale included questions about how respondents think their supplier understands the information they share; is honest; has sufficient human and material resources; can deliver a competent quantity and quality of the product; is ready to help the manufacturer; will consider the manufacturer’s interests when making decisions; and shares the manufacturer’s goals. For measuring reliability, Cronbach’s alpha value of the TRS scale was 0.952.

#### 3.3.2. Manufacturer Resilience

Manufacturer Resilience consists of three sub-variables. First, PPA consists of six statements from [88,89]. These items show the realities of the manufacturing companies where the respondents work before emergencies. Specifically, the items on the scale included questions about the extent to which the firm can identify and eliminate controllable risks in advance; maintains safety stocks and buffer stocks; can keep inventory levels and customer demand levels visible; has the personnel to monitor risks to the production process; has supply stocks and personnel trained to deal with supply or production disruptions; and has contingency plans in place based on experience and knowledge. Cronbach’s alpha coefficient of the PPA scale was 0.931.

Second, RPA consists of seven statements from [86,87,88,90,91,92]. The questions reflect the realities of a manufacturing company in emergencies. Specifically, the questions in the scale reflect the extent to which manufacturing companies can flexibly adapt internal and external workflows; implement contingency plans quickly; react quickly to repurpose resources; keep organizational structures and production stable; increase or decrease the number of suppliers reasonably; find the root cause of disruptions; and identify new opportunities and risks based on existing knowledge. To validate the reliability of the RPA scale, a Cronbach’s alpha coefficient of 0.941 was calculated.

Third, RCA consists of seven statements from [8,90,91,92]. The scale reflects the actual situation of a manufacturing company after emergencies have occurred. Specifically, the questions in the scale reflect the extent to which manufacturing companies can return to a new stable status; connect interrupted links quickly; restart production quickly; maintain essential functions in all departments; coordinate efforts to reduce the harm to the company from the contingency; and learn from experience to cope with future contingencies. Cronbach’s alpha coefficient of the RCA scale was 0.925.

#### 3.3.3. Collaborative Action (COL)

Collaborative action consists of nine statements from [33,61,86,93,94]. The scale reflects the extent to which manufacturing companies and suppliers collaborate in their actions before and after emergencies. Specifically, the questions in the scale reflect the extent to which the manufacturing company frequently discusses with suppliers the next phase of production volumes and types of products; conducts joint planning with suppliers to anticipate risks and problems in operations; frequently discusses with suppliers the contingency plans for product development and production; forecasts product demand with suppliers; shares long-term strategic plans for production with suppliers; resolves business issues and conflicts with suppliers; has a support team responsible for solving urgent problems; provides expertise or technology to complete tasks with suppliers; and shares responsibilities with suppliers. Cronbach’s alpha coefficient of the COL scale was 0.970.

#### 3.3.4. Environmental Uncertainty (ENU)

ENU consists of five statements modified from Wuyts and Geyskens’s [95] research, incorporating interviews with corporate experts. The scale reflects the uncertainty of the environment in which the company operates. Specifically, the questions in the scale reflect the extent to which manufacturing companies are unable to predict when emergencies will occur; find it challenging to anticipate market demand; have difficulty predicting competitors’ reactions; have difficulty implementing technological innovations that affect production; and cannot foresee whether they will survive in the marketplace in the long term. Cronbach’s alpha coefficient of the ENU scale was 0.830.

### 3.4. Model Assessment and Factor Analysis

The model assessment and factor analysis consist of three phases in this research. In the first stage, we tested the reliability and internal reliability of the scale, as shown in Table 3. The outer loadings of the items are all above 0.60, indicating that the questionnaire has acceptable reliability [96,97]. Additionally, the researchers tested Cronbach’s alpha and the composite reliability (CR) scores as internal consistency reliability measures. As shown in Table 3, for the TRS indicator, the CR equals 0.890, indicating decent internal consistency. Similarly, for the COL, PPA, RPA, and RCA indicator, the CR values are larger than 0.7, showing good internal consistency reliability for the measurement scales [98,99].

**Table 3 behavsci-13-00033-t003:** The constructs and measurement items.

Constructs and Items	Coding	Loadings	AVE	CR	Cronbach’s Alpha
Trust relationship with suppliers adapted from [28,86,87]	TRS				
The extent you think your supplier					
Understands the information we share	TRS1	0.755	0.537	0.890	0.952
Is honest to us	TRS2	0.771			
Has sufficient human and material resources	TRS3	0.727			
Can deliver a competent quantity and quality of the product	TRS4	0.669			
Is ready to help the manufacturer	TRS5	0.737			
Will consider the manufacturer’s interests when making decisions	TRS6	0.726			
Shares the manufacturer’s goals	TRS7	0.739			
Collaborative action adapted from [33,61,86,94]	COL				
To what extent your manufacturing company					
Frequently discusses with suppliers the next phase of production volumes and types of products	COL1	0.826	0.636	0.940	0.970
Conducts joint planning with suppliers to anticipate risks and problems in operations	COL2	0.812			
Frequently discusses with suppliers the contingency plans for product development and production	COL3	0.822			
Forecasts product demand with suppliers	COL4	0.790			
Shares our long-term strategic plan for production with suppliers	COL5	0.802			
Resolves business issues and conflicts with suppliers	COL6	0.781			
Has a support team responsible for solving urgent problems	COL7	0.809			
Provides expertise or technology to complete tasks with suppliers	COL8	0.802			
Shares responsibilities with suppliers	COL9	0.730			
Preparedness adapted from [88,89]	PPA				
To what extent your manufacturing company					
Can identify and eliminate controllable risks in advance	PPA1	0.788	0.616	0.906	0.931
Maintains safety stocks and buffer stocks	PPA2	0.794			
Can keep inventory levels and customer demand levels visible	PPA3	0.803			
Has the personnel to monitor risks to the production process	PPA4	0.806			
Has supply stocks and personnel trained to deal with supply or production disruptions	PPA5	0.800			
Has contingency plans in place based on experience and knowledge	PPA6	0.716			
Responsiveness adapted from [86,87,88,90,91,92]	RPA				
To what extent your manufacturing company					
Can flexibly adapt internal and external workflows	RPA1	0.826	0.658	0.931	0.941
Implement contingency plans quickly	RPA2	0.827			
React quickly to repurpose resources	RPA3	0.800			
Keep organizational structures and production stable	RPA4	0.820			
Increase or decrease the number of suppliers reasonably	RPA	0.800			
Find the root cause of disruptions	RPA6	0.780			
Identify new opportunities and risks based on existing knowledge	RPA7	0.822			
Recovery capability adapted from [8,90,91,92]	RCA				
To what extent your manufacturing company					
Can return to a new stable status	RCA1	0.715	0.558	0.883	0.925
Connect interrupted links quickly	RCA2	0.769			
Restart production quickly	RCA3	0.725			
Maintain essential functions in all departments	RCA4	0.779			
Coordinate efforts to reduce the harm to the company from the contingency	RCA5	0.764			
Learn from experience to cope with future contingencies	RCA6	0.728			
Environmental uncertainty adapted from [95]	ENU				
To what extent your manufacturing company					
Is unable to predict when emergencies will occur	ENU1	0.731	0.561	0.865	0.830
Finds it challenging to anticipate market demand	ENU2	0.752			
Has difficulty predicting competitors’ reactions	ENU3	0.766			
Has difficulty implementing technological innovations that affect production	ENU4	0.774			
Cannot foresee whether they will survive in the marketplace in the long term	ENU5	0.720			

Note: TRS: trust relationship with suppliers; COL: collaborative action; PPA: preparedness; RPA: responsiveness; RCA: recovery capability; ENU: environmental uncertainty; AVE: average variance extracted; CR: composite reliability.

In the second stage, as the scales used in this paper are not identical to those used previously, exploratory factor analysis was undertaken to validate the demarcation of the scale’s dimensions (see Table 4). The Kaiser–Meyer–Olkin (KMO) value was 0.967, indicating that the data can be used for exploratory factor analysis.

In the third stage, to evaluate the convergent validity of the constructs, the researchers employed confirmatory factor analysis (CFA). As shown in Table 3, the average variance extracted (AVE) values all surpass 0.5 [99], indicating that each concept explains more than 50% of the variance among the elements [98]. In this way, the results illustrate the acceptability of the constructs’ validity (see Table 2). The findings of the discriminant validity tests are provided in Table 5, in which the researchers compared the square root of the AVE of each construct to its correlation. The findings indicate that the square roots of the AVE for all constructs were larger than their respective correlation values. This indicates that the constructs in the scale utilized in this investigation have appropriate discriminant validity [98].

After checking the scales’ quality, we demonstrated the suitability of the research model for structural equation analysis. We tested the model fit of the structural equation model [100]. The results showed that the model of this study was suitable for structural equation analysis (see Table 6).

## 4. Results

### 4.1. Direct Effects

This study used a structural equation model (SEM) to test the hypothesis of each direct effect. For testing the H1 hypothesis, the results confirmed a positive effect (β = 0.702) between TRS and COL at *p* < 0.001 (see Table 7 and Figure 2). This means that H1 was empirically maintained. In addition, we examined the direct effects of COL on PPA, RPA, and RCA. For testing the H2a hypothesis, the β value of the “COL to PPA” path was 0.228 at a significance level of *p* < 0.001. Thus, hypothesis H2a was supported, indicating a significant correlation between COL and PPA. This means that COL has a significant positive effect on PPA. Then, for testing the H3a hypothesis, the β value was 0.352 of the path “COL to RPA” at a significance level of *p* < 0.001. Therefore, COL had a positive impact on RPA, as hypothesis 3a suggested. Finally, for the H4a hypothesis, the β value was 0.260 of the path “COL to RCA” at *p* < 0.001. Thus, H4a was supported, showing a positive correlation between COL and RCA.

### 4.2. Mediation Analysis

The researchers performed a mediating effect analysis to assess the mediating role of COL on the linkage between TRS and MR in the preparing, response, and recovery phases [75]. The results of the total, direct, and mediation effect revealed the following.

First, H2b proposed that COL mediated the relationship between TRS and PPA. The bias-corrected confidence interval (CI) (0.087 to 0.246) and the percentile CI (0.083 to 0.242) of the indirect effect exclude zero. Additionally, the two kinds of CI of the direct effect were (0.342 to 0.557) and (0.344 to 0.559), both of which exclude zero. In addition, the two kinds of confidence intervals of the total effect were (0.551 to 0.676) and (0.551 to 0.676), both of which exclude zero. Therefore, the results show that H2b was supported, indicating the partial mediation effect of TRS on PPA through COL was found significant [101].

Second, H3b hypothesized that COL mediated the relationship between TRS and RPA. The two kinds of CI of the indirect effect were (0.143 to 0.364) and (0.143 to 0.364); both excluded zero. The two kinds of CI of the direct effect were (0.086 to 0.385) and (0.085 to 0.383); both of them excluded zero. In addition, the two kinds of CI of the total effect were (0.395 to 0.574) and (0.396 to 0.575), both of which excluded zero. Consequently, results show that H3b was supported, indicating that H3b was supported and the partial mediation effect of TRS on RPA through COL was significant.

Third, for hypothesis H4b, the two kinds of CI of the indirect effect were (0.096 to 0.282) and (0.096 to 0.283); both excluded zero. The two types of CI of the direct effect were (0.393 to 0.628) and (0.389 to 0.625); both excluded zero. In addition, the two kinds of CI of the total effect were (0.610 to 0.770) and (0.614 to 0.773); both excluded zero.

We found a partial mediation for the association among TRS on RCA via COL. As shown in Table 8, the result revealed that the mediating effect was significantly positive, suggesting that H4b was supported.

Furthermore, this study conducted bootstrapped mediation analyses to evaluate whether the results were robust. Through Bootstrap 5000, all results of the mediating effect were significant, and none of the confidence intervals for the direct, indirect, and total effects contained 0.

### 4.3. Moderating Effect

A moderating analysis was performed to assess environmental uncertainty’s moderate effect on the association between TRS and COL. Table 9 demonstrates the results. Hypothesis 5 sought to ascertain ENU’s moderating role among TRS and COL. The results indicated that ENU does not moderate the association between TRS and COL (Beta = −0.0515 at a significance level of *p* = 0.315, t = −2.143). ENU does not have a moderating effect.

## 5. Discussion and Implications

### 5.1. Discussion

This study aims to empirically examine the association between TRS, collaborative action, and manufacturer resilience according to the PPRR theory. Additionally, it tests the moderating effect of environmental uncertainty on the correlation between TRS and collaborative action, with the application in the manufacturing sector. This research uses the knowledge of behavior operation management to solve the risk management problem in manufacturing enterprises. Built on a sample of 382 managers in manufacturing firms and independent directors, we found that TRS positively affects collaborative action. Based on the trust theory, this result is similar to the research indicating that trust is an implicit condition for collaboration [41,52]. Moreover, the results of testing H2a, H3a, and H4a reveal that COL positively impacts manufacturer resilience (PPA, RPA, and RCA). These results are similar to the studies pointing out that closer collaborative action between suppliers and manufacturers helps to share risks and increase productivity and resilience over time [37,55]. Specifically, according to the synergy theory, the findings are consistent with prior assertions that supply chain cooperation resulted in enhanced business performance [57]. Polyvio et al. [12] studied the impact of relational capital on the resilience of firms facing disruptions, which is consistent with the results of this study. That is, social capital exchanged by firms in cooperation enhances firm coordination and reduces uncertainty, and thus increases resilience.

Furthermore, our results of H2b, H3b, and H4b reveal that collaborative action partially mediates the relationships between TRS and PPA, RPA, and RCA. Additionally, studies such as the one performed by de Paula et al. [66] mentioned that specific collaborative activities between manufacturers and suppliers, such as collaboration and collaborative relationship efforts, can improve a company’s visibility, speed, and flexibility. In the same vein, Mcevily and Marcus [61] pointed out that joint problem solving between firms can facilitate knowledge transfer between firms and reduce opportunistic behavior through frequent interactions. These findings have built an unprecedented bridge between supplier relationship governance and manufacturer resilience. Using multidisciplinary knowledge and based on the research of Yang et al. [94], we developed a collaborative action scale to quantify the joint planning and emergency response between suppliers and manufacturers in the context of emergencies. This contributes to the literature on relationship management.

Surprisingly, the results of H5 do not verify the negative moderating impact of ENU on the association between TRS and collaborative action. This insignificant effect is attributed to the following reasons. First, according to Campos et al. [41], trust is a fundamental collaboration factor. Additionally, some argue that trust is the basis for collaborative action. The reluctance to act collaboratively is due to a lack of established trust. For instance, trust relationships are vital in shaping new supplier–manufacturer relationships during supply chain collaboration. Consequently, trust will certainly facilitate collaborative action, whatever the environment [47]. Second, companies with cooperative relationships have trusting relationships but are playing games, so the uncertain environment inhibits their collaborative actions [95]. Furthermore, influenced by Chinese culture, the crisis environment has contributed even more to the national cohesion of companies while increasing the synergy of action between them. Chinese culture embraces patriotism and solidarity. The uncertain environment motivates companies to be patriotic and act in concert; they are more willing to sacrifice their interests to help others and unite other companies to enhance collective action. This promotes corporate collaboration based on trust and reflects the spirit of solidarity and mutual brotherhood. This phenomenon is in line with traditional Chinese culture. Thus, the effect of environmental uncertainty on the trust–collaborative action pathway is insignificant in terms of its dissipative effect under the influence of multiple effects. Finally, during a pandemic, environmental uncertainties such as supply uncertainty and logistical uncertainty make people more concerned about compromising their interests when acting jointly. Uncertain environments are more likely to breed opportunistic or self-protective behaviors by firms to guard against risk, which inhibits the contribution of trust to collaborative action [94]. On the other hand, a moderate level of trust helps to prevent opportunistic behavior. It is therefore not possible to determine, in an environment of great uncertainty during a pandemic, whether there is more trust in preventing opportunistic behavior, or more opportunistic behavior generated in joint operations. A collaborative relationship between supplier and manufacturer does not mean they have the same legal basis as a formal partnership structure [47]. In this regard, manufacturing companies can persuade suppliers to support the resilience-enhancing effects of relationship governance by establishing extra-contractual governance provisions, or by adding synergistic actions to the terms of the contract [35].

### 5.2. Theoretical Contributions

This study makes the following academic contributions to the existing literature. Firstly, this research adds to the supplier relationship management theory. The previous research is limited as it only focuses on supply chain resilience in a normal state [4]. This study uses empirical methods to investigate the actual situation of companies under the pandemic’s influence, which extends the previous work [1,3,4]. This study integrates psychology, disaster science, game theory, and emergency management knowledge. Additionally, it brings behavioral operations management theories into the risk management area, showing newly discovered findings of how collaborative action can mediate TRS and resilience during the pandemic, based on the previous studies predominantly focusing on the whole supply chain resilience [1]. This paper gives a comprehensive investigation of the relationship between manufacturers and suppliers in the supply chain, which makes the research questions more relevant and comprehensive. The above theoretical contributions are useful for manufacturing companies to propose emergency policies for collaborative risk management with suppliers.

Secondly, this study contributes to the risk management theory. We use Chinese methods to solve Chinese problems. Influenced by traditional Chinese Confucianism, managers value non-contractual provisions for trust and solidarity, and contractual governance is not fully adequate to effectively manage partnerships [35]. Therefore, in this study, collaborative actions in the context of emergencies encompass joint contingency planning and joint problem solving. These practices are contributing to risk management theory in specific ways [15]. This paper is based on the theory of collaborative risk management, and incorporates ideas from the Chinese scholar Sun Tzu’s *Art of War* for the first time [26]. This facilitates emergency management for Chinese manufacturing companies in the event of an emergency [102]. Furthermore, this study enriches the context of the application of trust theory and synergy theory. The importance of relationships in shaping supply chain collaboration is based on the trust theory; that is, the positive belief, attitude, or expectation of one party that the actions or outcomes of the other party are likely to be satisfactory [17]. We focus on manufacturing firms and demonstrate the role of collaborative action on resilience and the mediating role of collaborative action between trust and resilience. In addition, through literature analysis and surveys of manufacturing firms, this study provides specific approaches of collaborative action in contingency situations to provide a scale and theory for future research on collaborative action in emergencies. The scale integrates scales such as supply chain synergy and joint action, and can be used by future researchers in the field of risk management [22,103]. This study innovatively bridges collaborative emergency action and risk management for manufacturers.

Finally, we investigate how an unpredictable environment moderates the link between two kinds of relational governance, trust, and collaborative action. Interestingly, the results are inconsistent with the hypothesis, and we find that the facilitation effect of trust on collaborative emergency action is not affected by environmental uncertainty. This reveals that the environment is conflicted and complex. These valuable theoretical implications align with the Chinese context and reality. Therefore, this finding provides a new perspective for studying Chinese supplier-relational governance practices in different contexts.

### 5.3. Practical Implications

This study provides the following practical implications for managers of manufacturing firms. First, adding to the study of Dubey et al. [8], our study recommended that managers can be proactive in building trusting relationships with suppliers. Managers should be more cognizant of the significance of public and private networking, togetherness, trust, and conventions of cooperation, as well as the environment in which these might flourish, so they can construct more resilient supply chains [104]. It is worth noting that, for technologically advantaged suppliers, there are concerns about joint knowledge creation and intellectual property conflicts in their joint actions. Therefore, this study has inspired managers to act cooperatively to alleviate suppliers’ concerns about knowledge and skill theft only by establishing a trust relationship [42].

Secondly, agreeing with [22], the resilience that comes from collaborative action stems from, for instance, sharing knowledge and expertise, relationships of trust based on friendship, and formal (e.g., laws) and informal institutions (e.g., cultural norms and customs). Therefore, managers should involve suppliers in the collaborative planning and problem-solving process, and make them aware of the resilience-enhancing effects of partners working together to develop contingency policies at different manufacturing processes, such as inventory management [105]. Moreover, supporting [2], managers should develop intelligent solutions that increase trust and workflow transparency with partners. Meanwhile, managers should make sure that employees fully understand what the different stages of resilience entail, and what collaborative action involves, so that responsibility is assigned to different departments and the company has a high level of responsiveness. This paper calls for managers to act in concert with suppliers in these uncertain times with the goal of resilience, not just profit, to enhance the long-term interests of the partner company. For example, partners should update their previous zero-inventory strategy and maintain redundant inventory in the collaborative plan to protect against disruption risk.

Thirdly, the scale we developed reflects the extent to which manufacturing companies and suppliers collaborate in their actions before and after emergencies. The research’s results are consistent with Dubey et al. [16], which shows that managers should attach importance to the coordinated response of different stages of emergencies. With regard to collaborative planning, the manager of a manufacturing company should enact the following: discuss frequently with suppliers the next phase of production volumes and types of products; conduct joint planning with suppliers to anticipate risks and problems in operations; discuss contingency plans for product development and production; forecast product demand with suppliers; and share long-term strategic plans for production with suppliers. For example, at monthly, quarterly, or annual meetings between manufacturers and suppliers, they can plan long-term emergency strategies and discuss them together. Managers can refer to the contents of this paper’s scale, which can provide some insight and actionable advice for company operations. In terms of collaborative problem solving, the manager of a manufacturing company should enact the following: resolve business issues and conflicts with suppliers; set up a support team to address urgent problems; provide expertise or technology to complete tasks with suppliers; and share responsibilities with suppliers.

It is worth noting for managers that conflict in the relationship is inevitable throughout the collaborative action process. Managers need to facilitate contact and communication between manufacturers and suppliers properly, avoiding information barriers and developing sensible solutions with resilience as the goal [47]. Moreover, the results show that environmental uncertainty does not moderate the promoting effect of trust on collaborative action. Managers should build trust first, focusing on collaborative planning and collaborative problem solving. This is because trust facilitates collaborative action regardless of the environment [17].

The findings of this paper can be supported by real-life business practices. For example, when we investigated a food manufacturing company, the manager told us that due to supply disruptions brought about by the pandemic, ingredients from abroad could not arrive and they had to build trust with a new local supplier for a short time. Manufacturers and suppliers quickly established partnerships, streamlined contracting processes such as inspections and authorizations, and worked together to solve supply disruptions with a win–win goal. As another example, BYD Auto, a major car manufacturer, switched production to masks during the pandemic with no production experience. They found partner suppliers across sectors in a short period, responded quickly based on trust and synergy, and upgraded their original cooperation model in new areas to ensure the resilience of their mask production line. In addition, the garment manufacturing company, Jihua, began to link up internally and externally after the pandemic outbreak. They worked with several new suppliers and ordered large quantities, producing masks and protective clothing day and night to overcome the shortage of raw materials.

Furthermore, due to inadequate logistics, customs restrictions, and material shortages, similar stories exist in various Chinese provinces for manufacturing companies in various industries. Such companies include electronic product manufacturers such as Wuhan Hongxin, metal product manufacturers in Hubei Province, computer and electronic equipment manufacturers in Guangdong Province, chemical manufacturing and textile manufacturing companies in Zhejiang Province, and machinery manufacturers and biopharmaceutical companies such as Qinchuan Group in Shaanxi Province. We can see that these companies have improved the preparedness, responsiveness, and recovery capabilities of their manufacturing companies through collaborative action with their suppliers based on trust in the event of an emergency, guaranteeing the normal functioning of production tasks and supply chains. Through these real-life examples of typical companies, we can demonstrate that the theoretical model in this paper can be translated into real-life practice for many Chinese manufacturing companies. We also suggest that managers of companies can adopt the more specific collaborative action practices in the scale of this study in future emergencies to systematically enhance manufacturer resilience at different stages.

## 6. Conclusions and Future Directions

This study interviewed Chinese manufacturers and drew the following conclusions based on the trust theory and synergy theory. Firstly, trust has a positive impact on collaborative action. Secondly, collaborative action acts as a mediator between trust and PPA, RPA, and RCA. Furthermore, this study found that the facilitative effect of trust on collaborative action is not moderated by environmental uncertainty. This paper hopes to expand the scope of the application of the supplier relationship governance theory. In addition, this study calls on manufacturing managers to build trust and act in concert with suppliers with the goal of resilience, to improve the company’s responsiveness during emergencies such as COVID-19. The advantages of this paper are that, on the one hand, this study utilizes multidisciplinary knowledge to address risk management issues, making the research comprehensive. On the other hand, the empirical research we conducted during the pandemic was broad in scope and yielded a large amount of representative corporate data. This study draws on the well-established synergy theory to analyze the current practical situation, and provides new ideas for solving Chinese problems using Chinese methods.

Despite the increased focus on resilience among companies at risk of disruption during the pandemic, it is worth examining whether the impact of collaborative action on the resilience of manufacturing companies in the post-pandemic era can be sustained in the long term. In addition, our work has significant shortcomings that provide opportunity for further investigation. First, this research collected data on Chinese manufacturing companies during the COVID-19 pandemic in 2022, representing the situation of companies during this outbreak. Future research could develop longitudinal case studies to demonstrate companies’ specific use of relationship governance tools. Secondly, this study focuses on manufacturing companies and is not representative of the characteristics of suppliers or retailers. Future research could be carried out on different types of supply chain members. In addition, this study combines Chinese culture and a particular model of relational governance, and future research could explore the applicability of this approach to relational governance in other countries and regions. Finally, this study examines the effect of collaborative action as a mediating variable on manufacturers’ resilience, and future studies can explore the mediation or moderating mechanisms of other variables.

## Figures and Tables

**Figure 1 behavsci-13-00033-f001:**
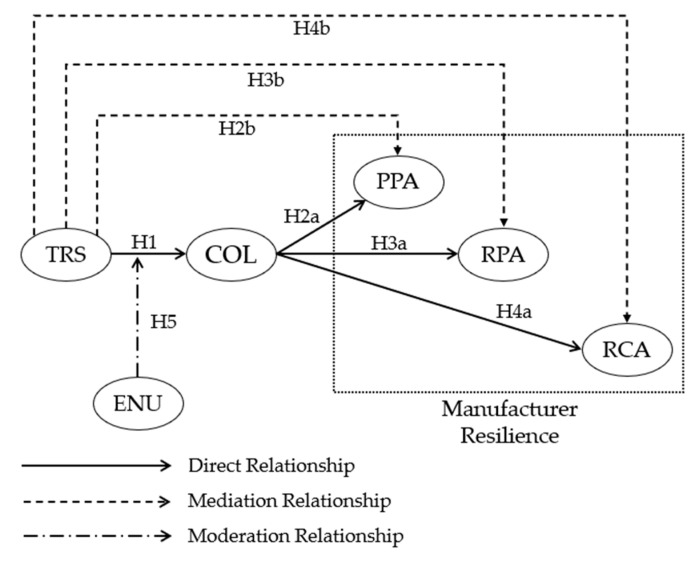
Theoretical model.

**Figure 2 behavsci-13-00033-f002:**
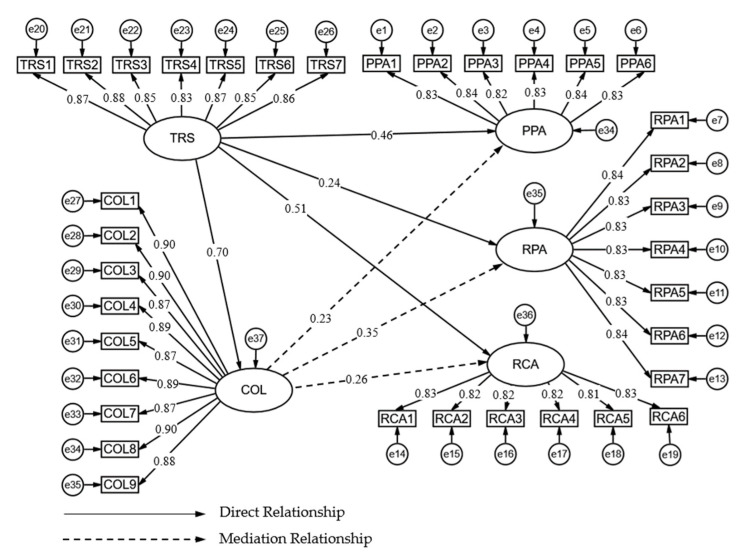
Structure Equation Model.

**Table 1 behavsci-13-00033-t001:** Descriptive statistics.

	Constructs and Items	Frequencies (382)	Percentage
Nature of enterprises	State-owned or state-owned holding	135	35.340%
	Private enterprise	188	49.215%
	Foreign-owned or Sino-foreign joint ventures	31	8.115%
	Other	28	7.330%
Industry type	Food and beverage manufacturing industry	51	13.351%
	Metallurgical manufacturing and processing/mechanical and equipment manufacturing industry	55	14.398%
	Pharmaceutical/chemical products manufacturing industry	55	14.398%
	Textile and clothing manufacturing industry	38	9.948%
	Wood furniture/sports goods manufacturing industry	34	8.901%
	Manufacturing industry of communications equipment, computers, and other electronic equipment	43	11.257%
	Others	106	27.749%
Enterprise size (number of employees)	1–50	25	7%
	51–300	45	13%
	301–2000	138	39%
	>2001	143	41%
Enterprise age	<5	23	6.021%
	5–10	20	5.236%
	11–15	37	9.686%
	>15	302	79.058%

**Table 2 behavsci-13-00033-t002:** Common Method Bias Test.

	χ^2^/df	RMSEA	CFI	IFI	TLI
Original Model	1.388	0.032	0.983	0.983	0.982
Single-Factor Model	9.956	0.153	0.602	0.603	0.578
Common Method Factor Model	1.233	0.025	0.990	0.991	0.989
Model Fit Variation		ΔRMSEA	ΔCFI	ΔIFI	ΔTLI
		0.007	0.007	0.008	0.007
Criteria		<0.05	<0.1	<0.1	<0.1

Note: RMSEA: root mean square error of approximation; CFI: comparative fit index; IFI: incremental fit index; TLI: Tucker–Lewis index.

**Table 4 behavsci-13-00033-t004:** Rotating component matrix.

Index	1	2	3	4	5	6
PPA1			0.788			
PPA2			0.794			
PPA3			0.803			
PPA4			0.806			
PPA5			0.800			
PPA6			0.716			
RPA1	0.826					
RPA2	0.827					
RPA3	0.800					
RPA4	0.820					
RPA5	0.800					
RPA6	0.780					
RPA7	0.822			0.715		
RCA1				0.769		
RCA2				0.725		
RCA3				0.779		
RCA4				0.764		
RCA5				0.728		
RCA6				0.715		
TRS1		0.755				
TRS2		0.771				
TRS3		0.727				
TRS4		0.669				
TRS5		0.737				
TRS6		0.726				
TRS7		0.739				
COL1					0.826	
COL2					0.812	
COL3					0.822	
COL4					0.790	
COL5					0.802	
COL6					0.781	
COL7					0.809	
COL8					0.802	
COL9					0.730	
ENU1						0.731
ENU2						0.752
ENU3						0.766
ENU4						0.774
ENU5						0.720

Note: TRS: trust relationship with suppliers; COL: collaborative action; PPA: preparedness; RPA: responsiveness; RCA: recovery capability; ENU: environmental uncertainty.

**Table 5 behavsci-13-00033-t005:** Correlations and discriminate validity.

Construct	Mean	SD	TRS	COL	PPA	RPA	RCA	ENU
TRS	4.789	1.540	**0.733**					
COL	4.879	1.612	0.675	**0.798**				
PPA	3.452	1.412	0.581	0.523	**0.785**			
RPA	5.858	1.119	0.459	0.494	0.283	**0.811**		
RCA	4.715	1.033	0.649	0.584	0.431	0.503	**0.747**	
ENU	3.904	1.090	−0.213	−0.282	−0.395	−0.135	−0.213	**0.749**

Note: All the correlations were significant at *p* < 0.01. Bold values in the diagonal are the square roots of average variance extracted from the constructs. TRS: trust relationship with suppliers; COL: collaborative action; PPA: preparedness; RPA: responsiveness; RCA: recovery capability; ENU: environmental uncertainty; SD: standard deviation.

**Table 6 behavsci-13-00033-t006:** Results of model fit.

Index	χ^2^/df	RMSEA	CFI	RFI	NFI	IFI	TLI	SRMR
Criteria	<3	<0.1	>0.9	>0.9	>0.9	>0.9	>0.9	<0.08
Result	1.422	0.033	0.981	0.936	0.94	0.982	0.98	0.047

Note: RMSEA: root mean square error of approximation; CFI: comparative fit index; RFI: relative fit index; NFI: normed fit index; IFI: incremental fit index; TLI: Tucker–Lewis index; SRMR: standardized root mean square residual.

**Table 7 behavsci-13-00033-t007:** Path relationships of the direct effects.

Hypothesis	Path	Std.Beta	SE	*p*	Results
H1	TRS ⟶COL	0.702	0.05	***	Supported
H2a	COL⟶PPA	0.228	0.054	***	Supported
H3a	COL⟶RPA	0.352	0.047	***	Supported
H4a	COL⟶RCA	0.260	0.037	***	Supported

Note: *p*: significance level; *** *p* < 0.001.TRS: trust relationship with suppliers; COL: collaborative action; PPA: preparedness; RPA: responsiveness; RCA: recovery capability; SE: standard error.

**Table 8 behavsci-13-00033-t008:** Mediation effect results.

					Bootstrapping			
					Bias-Corrected		Percentile	
					95% CI		95% CI	
Hypothesis		Estimate	SE	Z	Lower	Upper	Lower	Upper
H2b: TRS⟶COL⟶PPA	Indirect effect	0.160	0.040	4.000	0.087	0.246	0.083	0.242
	Direct effect	0.455	0.054	8.426	0.342	0.557	0.344	0.559
	Total effect	0.615	0.032	19.219	0.551	0.676	0.551	0.676
H3b: TRS⟶COL⟶ RPA	Indirect effect	0.247	0.056	4.411	0.143	0.364	0.143	0.364
	Direct effect	0.239	0.077	3.104	0.086	0.385	0.085	0.383
	Total effect	0.486	0.046	10.565	0.395	0.574	0.396	0.575
H4b: TRS⟶COL⟶RCA	Indirect effect	0.183	0.048	3.813	0.096	0.282	0.096	0.283
	Direct effect	0.514	0.061	8.426	0.393	0.628	0.389	0.625
	Total effect	0.697	0.041	17.000	0.610	0.770	0.614	0.773

Note: TRS: trust relationship with suppliers; COL: collaborative action; PPA: preparedness; RPA: responsiveness; RCA: recovery capability; SE: standard error; CI: confidence interval.

**Table 9 behavsci-13-00033-t009:** Moderating effect results.

	Beta	T Value	Sig.	Beta	T Value	Sig.	Beta	T Value	Sig.
Nature	−0.017	−0.345	0.730	−0.028	−0.743	0.458	−0.026	−0.682	0.496
Industry	0.141	2.794	0.005	0.064	1.678	0.094	0.061	1.587	0.113
Size	−0.247	−4.068	0.000	−0.109	−2.341	0.020	−0.108	−2.326	0.021
Age	0.071	1.158	0.248	0.017	0.377	0.707	0.013	0.283	0.777
TRS				0.633	15.902	0.000	0.641	15.809	0.000
ENU				−0.061	−1.519	0.130	−0.061	−1.517	0.130
TRS × ENU							−0.039	−1.006	0.315
R^2^		0.069			0.690			0.691	
F		6.933			56.789			48.823	

Note: Sig.: significance level; TRS: trust relationship with suppliers; ENU: environmental uncertainty.

## Data Availability

The data presented in this study are available on request from the corresponding author. The data are not publicly available due to respondents’ requests.

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
