# Peer review of "Trust Relationship with Suppliers, Collaborative Action, and Manufacturer Resilience in the COVID-19 Crisis"

_behavsci, 2022, doi:10.3390/bs13010033_

Round 1
Reviewer 1 Report
Trust Relationship with Suppliers, Collaborative Action and Manufacturer Resilience in the COVID-19 Crisis
1. Data Collection
- The sample of Chinese manufacturing companies in this study was drawn from the following sources: members of the China Association for Quality, members of the China 340 Federation of Logistics and Purchasing, manufacturing companies with which the re-341 search team collaborated on research projects, members of independent director training 342 courses of listed companies and students of MBA classes in Chinese universities
à So what is the population of this study?
- The data 343 were obtained between January 2022 to September 2022: For cross-sectional study, this period is too long. Please bring evidence to protect this aspect of the paper.
- Before the questionnaire was for-344 mally distributed, we consulted with 12 academics in economics and management-related 345 professions and managers working in manufacturing companies to ensure that the ques-346 tionnaire was clear and understandable: Please provide evidence to support the choosen of 12 academics here. Why not 5 or 20 but 12?
- A total of 382 effective questionnaires were returned, and the response rate was 351 68.71%: How can you contact these companies. How can you keep the response rate of this study (which is very high)?
Please give detail discussion!
- Factors and Indicators: Please provide the detailed indicators for each factor. How can you identify these factors?
Please give a table with references for the indicators and factors
- Questionnaire: Please add the real questionnaire survey in the Appendix
- Doing factor analysis in SEM makes no sense. You build theoretical framework before running SEM. It means the factors are identified before implementing SEM (Figure 1. Theoretical model).
Many papers are facing with this very simple problem!
- Discussion: Practical discussions need to be improved. Please bring more practical and quantitative evidences to support your argument. For example, the volume of the market,…
In other words, you need to translate your theoretical framework to the real practice. And How your findings is supported by the real world (i.e., China)
Reviewer 2 Report
A few questions / comments and suggestions:
In Line 200, full term to define “PPRR theory”.
In Line 543-544, confusing for the head “3. Results” and subheading “3.1 Discussion”.
In Line 559-561, relevant to the study is not clear.
In Line 577-578, relevant to the study is not clear.
In Line 585-586, how to elaborate increasing the synergy of action between them, relevant to the study is not clear.
In Line 585-586, how to elaborate increasing the synergy of action between them, relevant to the study is not clear.
In Line 588-590, the role of the environment, what factors affect by the environment during this pandemic, relevant to the study is not clear.
In Line 617-619, relevance to the study is not clear.
In Line 626-629, how their role in mediating the relationship, relevance to the study is not clear.
In Line 675-677, how to share long-term strategic plans for production with suppliers, relevance to the study is not clear.
Round 2
Reviewer 1 Report
Thanks for your works. Here are my comments:
-"The population surveyed in this study were managers and independent directors of manufacturing companies" --> Clarify the number of population (i.e., 2500).
From this number of population, you can establish the sample needed for this study.
- The data 343 were obtained between January 2022 to September 2022: The authors bring evidence of how they did. But they did not answer my question, which is "For cross-sectional study, this period is too long. Please bring evidence to protect this aspect of the paper."
Please bring scientific evidence to demonstrate you are doing the right thing (i.e., methodology from previous studies). It is different from showing you are really do it.
- Please provide evidence to support the choosen of 12 academics here. Why not 5 or 20 but 12?
The answer of the authors are nonsense and did not focus on the question!\
- More evidences are required to translate your theoretical framework to the real practice. And How your findings is supported by the real world (i.e., China)
Round 3
Reviewer 1 Report
Dear the authors,
Please keep your work slowly and carefully. Your fast response make me very suprise. However, the quality is terribly low. If your submission is not in MDPI, this paper will be rejected with such low quality.
Please give more attention on three questions as follows:
- The number of population in this study was 556: How do you know this number. Please give evidence.
- We can see that some of the surveys lasted a year, some six months, some three months: Please give some citations for this sentence.
- Please provide evidence to support the choosen of 12 academics here. Why not 5 or 20 but 12?: Please investigate more previous studies towards this matter to provide a more meaningful explanation with scientific evidences. This explanation is too simple and self-judgement.
